# Development of a High-Throughput Calcium Mobilization Assay for CCR6 Receptor Coupled to Hydrolase Activity Readout

**DOI:** 10.3390/biomedicines10020422

**Published:** 2022-02-10

**Authors:** Sara Gómez-Melero, Fé Isabel García-Maceira, Tania García-Maceira, Verónica Luna-Guerrero, Gracia Montero-Peñalvo, Javier Caballero-Villarraso, Isaac Túnez, Elier Paz-Rojas

**Affiliations:** 1Canvax Biotech, Parque Científico y Tecnológico Rabanales 21, c/Astrónoma Cecilia Payne s/n, Edificio Canvax, 14014 Córdoba, Spain; fi.garciamaceira@gmail.com (F.I.G.-M.); tagama0802@gmail.com (T.G.-M.); veronicalunaguerrero@gmail.com (V.L.-G.); gracia7432@gmail.com (G.M.-P.); 2Department of Biochemistry and Molecular Biology, Faculty of Medicine and Nursing, University of Córdoba, Avda. Menéndez Pidal s/n, 14004 Córdoba, Spain; bc2cavij@uco.es (J.C.-V.); itunez@uco.es (I.T.); 3Maimonides Biomedical Research Institute of Cordoba, Avda. Menéndez Pidal s/n, 14004, Córdoba, Spain; 4Multiplex Biopharma, Parque Científico Rabanales 21, Calle Astronoma Cecilia Payne, 14014 Córdoba, Spain; pazrojaselier@gmail.com

**Keywords:** high-throughput screening, HTS, CCR6, hexosaminidase, calcium mobilization, cell-sensor, GPCR

## Abstract

CCR6 is a chemokine receptor highly implicated in inflammatory diseases and could be a potential therapeutic target; however, no therapeutic agents targeting CCR6 have progressed into clinical evaluation. Development of a high-throughput screening assay for CCR6 should facilitate the identification of novel compounds against CCR6. To develop a cell-based assay, RBL-2H3 cells were transfected with plasmids encoding β-hexosaminidase and CCR6. Intracellular calcium mobilization of transfected cells was measured with a fluorescent substrate using the activity of released hexosaminidase as readout of the assay. This stable, transfected cell showed a specific signal to the background ratio of 19.1 with low variability of the signal along the time. The assay was validated and optimized for high-throughput screening. The cell-based calcium mobilization assay responded to the specific CCR6 ligand, CCL20, in a dose-dependent manner with an EC_50_ value of 10.72 nM. Furthermore, the assay was deemed robust and reproducible with a Z’ factor of 0.63 and a signal window of 7.75. We have established a cell-based high-throughput calcium mobilization assay for CCR6 receptor. This assay monitors calcium mobilization, due to CCR6h activation by CCL20, using hexosaminidase activity as readout. This assay was proved to be robust, easy to automate and could be used as method for screening of CCR6 modulators.

## 1. Introduction

G-protein-coupled receptors (GPCRs) represent 4% of the human genome and are one of the most attractive therapeutic targets due to their involvement in many important physiological functions [1]. Despite being one of the largest classes of proteins, the majority of GPCRs remains undragged, with only approximately 100 GPCRs targeted with small molecules or peptides [2]. GPCR receptors do not have activity per se, but they are linked to GTPases that participate in signal transduction. Such GTPases are collectively known as G proteins and each G protein comprises three subunits, α, β, and γ. There are four families of G proteins that are classified according to their α subunit: G_s_, G_i_, G_q_ and G_12/13_ where G_s_ and G_i_ participate in cAMP-dependent pathways, G_q_ activates phospholipase C-dependent pathways, and G_12/13_ activates a small family of GTPases [3,4,5]. In addition, the G_q_ family is further divided into four G_q_ proteins due to the use of four different α polypeptides, αq, α11, α14, and α15/16 [6]. The G_α15/16_ polypeptide has a restricted tissue expression and is predominantly expressed in myeloid and B-cell lineages while G_αq_ and G_α11_ are ubiquitously expressed [6].

There have been extensive efforts to develop new, sensitive and simple-to-use assays to measure second messengers or molecules coupled to the signal transduction as readout for GPCR activation [7]. Based on this signal transduction cascade several assay techniques such as radioligand binding, reporter gene, cAMP detection and calcium mobilization are commonly used [8]. Calcium mobilization is the result of activation of GPCRs through specific subunit G_αq_ activating classical phospholipase C pathways resulting in the release of calcium from the endoplasmic reticulum into the cytoplasm [7]. In resting cells, the cytosolic calcium concentration is lower (~100–200 nM) than that in the extracellular environment (~2 mM), but when the cells are excited by the activation of GPCRs the concentration of intracellular calcium can increase in seconds to 100 µM [9]. Because this change in the concentration of intracellular Ca^2+^ play a crucial role in GPCR-induced signaling, the develop of measurement assays based on GPCR calcium mobilization has attracted more attention in GPCR-targeted drug discovery and has widely used in the investigation of many GPCRs [10].

The human CC chemokine receptor 6, hCCR6, is a class A of GPCR expressed in a particular diverse range of leukocytes including T cells (specifically Th17 cells and Treg cells), B cell, neutrophils and subsets of dendritic cells [11]. With particular relevance in rheumatoid arthritis, psoriasis, multiple sclerosis, inflammatory bowel disease and cancer; inhibition of the hCCR6 signaling might prove to be a useful therapeutic strategy [11,12]. Some studies indicate that upon CCR6 binding to its ligand, CCL20 is activated a signaling pathway through calcium mobilization [13,14]. Therefore, the development of novel functional assays based on calcium mobilization is of interest for CCR6h-targeted drug discovery [10].

Extensive research has been conducted regarding the molecular determinants of coupling of each GPCR to a particular family of G proteins and today it is known that GPCR and G protein coupling selectivity involves the interaction of GPCR with both the extreme carboxyl terminus and linker I of G_α_ protein [15] and that development of chimeras of G_αq_ comprising the C-terminal five amino of G_αi_ or G_αs_ and a highly conserved glycine within Linker I is enough for the coupling of non-G_αq_ coupled GPCR to signaling through phospholipase C-dependent pathways [16]. In addition, it was previously known that G_α15/16_ is a promiscuous G protein, that is, it can couple to a wide variety of GPCR to signal through phospholipase C-dependent pathways [17] and this is the molecular basis of the measurement of calcium release upon activation of many non-naturally G_αq_ coupled-GPCRs assays that are used in HTS today; for example, the ChemiScreen™ family of HTS products RBL-2H3 is a rat basophilic leukemia cell line that expresses the G_α15/16_ of G_α_ subunit. This promiscuous G_α15/16_ protein shifts the receptors coupling to the calcium mobilization pathway [18]. Mast cells release their secretory granules after aggregation of their high affinity receptors, Fcε receptor I (FcεRI), by the antigen-immunoglobulin E (IgE) complexes [13,18]. Aggregation of FcεRI by antigens initiates a signaling cascade that induces the elevation of intracellular Ca^2+^ causing degranulation [18]. In this way, the FcεRI stimulation induces the release of hydrolases, such as β-hexosaminidase (HexB), from intracellular granules [13].

We hypothesized that overexpression of a GPCR receptor on the surface of RBL-HexB cells generates a cell-based sensor useful to determine calcium mobilization through GPCR ligand binding using hydrolase activity as readout of the assay. This assay can be used to measure the effect of compounds on calcium mobilization signaling of CCR6h receptor. When the cell-based sensor is incubated with the specific CCR6h ligand, CCL20, the amount of intracellular calcium increases, and the reporter polypeptide is released. The enzymatic activity of such released non-protease hydrolase reporter polypeptide is detected with the specific substrate. Herein we report the development of a precise, simple and cost-effective assay to measure calcium mobilization for the CCR6h receptor.

## 2. Materials and Methods

### 2.1. Plasmids

Coding sequence of human β-hexosaminidase (HexB, Genbank: BC017378.2) was synthesized by Integrated DNA Technologies (IDT), flanked with the recognition sites of 5’XhoI and 3’NotI and cloned into the pRV retroviral vector by the restriction with the indicated enzymes. A hygromycin resistance cassette was included in the vector backbone for stable cells selection. The vector also included an IRES-NGFR cassette, cloned downstream of human HexB and, thus, under the control of the same promoter, for selection by flow cytometry of cells expressing HexB stably.

Coding sequence of human CCR6 (Gb: NM_004367) was obtained by synthesis (IDT), flanked with the recognition sites of 5’XhoI and 3’NotI. It was cloned in frame with the signal peptide IgK, a short sequence (VGS) to improve GPCR expression [19] and a tag sequence (c-myc) into the pcDNA3.1 vector by the restriction with the indicated enzymes. pcDNA3.1 vector included a neomycin resistance cassette for stable cells selection.

The integrity of the constructs was confirmed by sequence analysis (Stabvida, Caparica, Portugal).

### 2.2. Cell Culture

All cell lines were purchased from the American Type Culture Collection (ATCC, Manassas, VA, USA). RBL-2H3 (rat basophilic leukemia, CRL-2256) cells was cultured in Eagle’s minimum essential medium (EMEM)/RPMI-1640 medium. MYC 1-9E10.2 (9E10, CRL-1729) and 200-6-G3-4 (20.4, HB-8737) hybridoma cells were maintained in RPMI-1640 medium. All media (Gibco) were supplemented with 10% fetal bovine serum (FBS, Gibco, New York, NY, USA), 2 mM L-glutamine (Lonza, Basel, Switzerland), 100 U/mL penicillin (Gibco), and 100 μg/mL streptomycin (Gibco). Cells were maintained at 37 °C in a humidified atmosphere of 5% CO_2_.

### 2.3. Antibody Purification

Mouse anti-c-myc and mouse anti-NGFR were obtained from culture supernatant of MYC 1-9E10.2 (9E10, CRL-1729) and 200-6-G3-4 (20.4, HB-8737) hybridoma cells, respectively. Immunoglobulins were affinity-purified from the culture supernatant using Protein G Sepharose columns (HiTrap Protein G HP, GE Healthcare, Madrid, Spain) according to the manufacturer’s protocol in AKTA Prime Plus (GE Healthcare). The purity of purified mAbs was checked by SDS-PAGE on 10% acrylamide gel and stained with Coomassie brilliant blue. The concentration of purified immunoglobulins was determined by UV absorbance at 280 nm.

### 2.4. Flow Cytometry

For positive transfected cells selection, flow cytometric analysis was performed. Mouse anti-c-myc (9E10, in house) or mouse anti-NGFR (HB-8737, in house) antibodies were incubated with RBL-2H3 transfected cells or mock transfectants for 20 min at 4 °C. After washing with PBS, cells were stained with FITC-conjugated goat anti-mouse IgG (1:60, Sigma, Burlington, MA, USA) for 10 min at 4 °C and were analyzed by flow cytometry using a FACSCalibur (Becton Dickinson, Madrid, Spain) cytometer. A total of 10,000 events were acquired and data were analyzed with BD CellQuest Pro software (BD CellQuest Pro, BD Biosciences, Madrid, Spain).

### 2.5. Creation of Stable RBL-2H3 Clone with β-hexosaminidase Activity and CCR6 Expression

RBL-2H3 was transfected, with 2 µg of the expression vector pRV-PCMV-HexB-IRES-NGFR-HYG obtained, using Canfast (Canvax Biotech, Córdoba, Spain) according to the manufacturer’s instructions. Following transfection, stably transfectants were selected with 2 mg/mL hygromycin B (Duchefa, Torrente, Spain). To confirm the plasmid transfection, flow cytometry against NGFR with mouse anti-NGFR (HB8737, in house) antibody was used. Positive cells were cloned by limiting dilution in 96-well plates and wells with growing colonies were analyzed for NGFR expression by flow cytometry again.

Clones with positive expression for NGFR were expanded to six well plates for determination of HexB activity. The cells were plated in 384-well plates (Corning, 384-well flat-bottom black polystyrene microplates, Corning, New York, USA) in 30 µL per well at 5000 cells per well and cultured for 48 h at 37 °C, 5% CO_2_. Then, cell media was aspirated and replaced with a mix containing 1 µg/mL mouse anti-TNP antibody (clone IgELb4, Creative Biolabs, Shirley, NY, USA), 1 µg/mL of TNP-BSA (Biosearch Technologies, Hoddesdond, UK) and 1 mM of 4-Methylumbelliferyl N-acetyl-β-D-glucosaminide (4MU-NGlc) substrate (Glycosynth) in BSS (25 mM HEPES/NaOH, 1.2 mM KH_2_PO_4_, 65 mM NaCl, 5.65 mM KCl, 0.6 mM MgCl_2_, 1.8 mM CaCl_2_, 5.6 mM glucose and 0.1% BSA). Cells were incubated at 37 °C protected from light. The fluorescence was measured at 360 nm excitation and 470 nm emission wavelength in a FLUOStar OPTIMA plate reader (BMG Labtech, Ortenberg, Germany) to measure exocytosis of HexB. A read was taken at 0, 15, 30, 45 and 60 min of incubation. To measure background release, 2 mM of 4MU-NGlc was used without anti-TNP antibody or TNP-BSA. Ionomycin (Sigma, Burlington, MA, USA) at 10 µM was used as a positive control. Wells without cells were used as blank and their fluorescence was subtracted of specific and background release to calculate the specific signal-to-background (S/B) ratio. The stability of HexB expression in 1B7-RBL-2H3-HexB cells’ activity in the time was measured for two months.

The selected clone, 1B7-RBL-2H3-HexB, with the best hexosaminidase activity was transfected with 2 µg of the expression vector pcDNA3.1-PEF1α-PSIgκ-CMyc-VGS-CCR6-Neo using Canfast (Canvax Biotech, Córdoba, Sapin) according to the manufacturer’s instructions. Following transfection, stably transfectants were selected with 0.5 mg/mL G418 (Phytotechnology, Lenexa, KS, USA). Flow cytometry was used to measure surface expression of CCR6 using an anti c-myc antibody (9E10 clone, mouse antibody) followed by anti-mouse-FITC (Sigma).

### 2.6. CCR6h Calcium Mobilization Assay

Mobilization of Ca^2+^ by CCL20 ligand was determined using 1B7-RBL-HexB-hCCR6 cells. The cells were plated in 384-well plates (Corning, 384-well flat-bottom black polystyrene microplates) in 30 µL per well at 5000 cells per well and cultured for 48 h at 37 °C, 5% CO_2_. Then, cell media was aspirated and replaced with serial dilutions of CCL20 (Peprotech, London, UK) and 1 mM 4-Mthylumbelliferyl N-acetyl-β-D-glucosaminide substrate (Sigma) per well in BSS protected from light. Cells were incubated 60 min at 37 °C, 5% CO_2_. The fluorescence was measured (excitation: 360 nm, emission: 470 nm) using a FLUOStar OPTIMA plate reader (BMG Labtech). To generate dose-response curves of CCL20 and calculate EC_50_ and Hill slope (H) value non-linear regression analyses were performed.

EC_50_ and H were used to calculate EC_80_ using the following equation:(1)$${\mathrm{EC}}_{F}={\left(\frac{F}{100-F}\right)}^{\frac{1}{H}}{*\mathrm{EC}}_{50}$$
where F is the fraction of maximal response, in this case 80.

### 2.7. Optimization and Performance of the HTS Assay

In order to optimize the incubation time, the assay was run as described above changing the incubation time with the ligand. Furthermore, the method of dispensing cells manually or using a Research Pro 8 Channel Electronic Multichannel Pipette (Eppendorf, Madrid, Spain) was compared. Concentration response curves of CCL20 were run to compare EC_50_ value and signal-to-background (S/B) ratio.

The assay with optimized conditions was performed to determine the Z’ factor and the signal window. The Z’ factor [20] was calculated by the following equation: Z’ = 1 − ((3SDmax + 3SDmin)/|mean max − mean min|) and the signal window (SW) [21] was calculated using the equation: SW = ((mean max − 3SDmax) − (mean min + 3SDmin))/SDmax. In both equations, SDmax is the standard deviation of the positive controls, SDmin is the standard deviation of the negative controls, mean max is the mean value of the positive controls and mean min is the mean value of the negative controls. CCL20 at 60 nM (approximately EC_80_) was used as a positive control and the assay buffer (BSS) as a negative control.

### 2.8. Statistical Analysis

Data were represented as mean ± SD. The statistical analysis was carried out using GraphPad Prism version 8.0.1 software (GraphPad Software Inc., San Diego, CA, USA). Non-linear regression analyses were performed to generate dose-response curves and calculate EC_50_ and EC_80_ values. Linear regression was used to analyze data reproducibility. Statistical significance was evaluated with Student’s *t*-test and are indicated with * *p* < 0.05; ** *p* < 0.01; *** *p* < 0.001; and **** *p* < 0.0001.

## 3. Results

### 3.1. Development of Stable Cell Line Expressing Human HexB

To obtain a cell line with hexosaminidase activity, we transfected RBL-2H3 cells with a plasmid for the expression of human HexB fused to IRES-NGFR cassette (Figure 1a). Transfected cells were kept in antibiotic selection with hygromycin for two weeks, after which, cells were tested in flow cytometry. Positive cells for NGFR expression were selected and cloned to ensure their stability and, growing colonies, were analyzed for NGFR expression again.

HexB activity of three clones with positive expression for NGFR, named 1B7-RBL-2H3-HexB, 1C4-RBL-2H3-HexB and 1F10-RBL-2H3-HexB, was assayed. We measured the release of hexosaminidase due to interactions between multimeric antigen and surface-bound IgE, as described in the Materials and Methods (Figure 1b). A read of fluorescence was taken at different times of incubation, selecting 60 min as the optimal assay time (Table 1).

The above results indicate that overexpression of HexB into RBL-2H3 cells produces a functional enzyme, as measured with 4MU-NGlc, that is stored inside the granules and is specifically released by exocytosis through cross-linking of IgE receptor. In the stably transfected population, RBL-2H3-HexB, the specific signal is increased 2.2 times with respect to mock transfectants, while background increased 1.4 times, that is, specific signal increased more than background release and this indicates that the uses of transfected cells as sensors is better than the use of untransfected cells. The specific signal to the background (S/B) ratio of untransfected RBL-2H3 cells in this experiment was 5.9, while the S/B ratio for RBL-2H3-HexB cells was 7.5. Clones selected by limiting dilution from the RBL-2H3-HexB increased specific release, but clones 1C4 and 1F10 also increased background release having a signal-to-background ratio of 7.4 and 5. However, the signal-to-background ratio of clone 1B7 was 19.1, which is an increment of 11.6 times of RBL-HexB ratio and 13.2 times of RBL untransfected cells ratio (Figure 2a). The clone 1B7-RBL-2H3-HexB, with the best signal-to-background ratio, was chosen as our working clone.

Even more important than the signal reached is the fact that production and release of hexosaminidase by the 1B7-RBL-2H3-HexB cells obtained is extremely regular while that of RBL-2H3 untransfected cells has a very strong variability over time. Both, RBL-2H3 and 1B7-RBL-2H3-HexB, cells were cultured for 2 months, and exocytosis was measured as above every 1 month. The signal-to-background ratio of RBL-2H3 cells was 5.9 (month 0), 2.8 (month 1) and 4.3 (month 2) while the signal-to-background ratio of 1B7-RBL-2H3-HexB cells was 19.1 (month 0), 17 (month 1) and 22 (month 2). A strong natural variability was observed in RBL-2H3 cells with a reduction of 63% of S/B ratio between month 0 and month 1. But 1B7-RBL-HexB cells behaved much better and the S/B ratio, while still variable as corresponds to live cells, was more stable and the maximal variability was 22% between month 1 and month 2 (Figure 2b).

### 3.2. Development of Calcium Mobilization Assay for CCR6h Receptor Coupled to β-Hexosaminidase Readout

To study CCR6 activation by CCL20 with β-hexosaminidase enzymatic assay, 1B7-RBL-2H3-HexB cells were transfected with an expression plasmid containing viral glycosylation sequence (VGS) and c-myc tag fused in frame upstream of CCR6h (Figure 3a). The VGS sequence was used to enhance CCR6h surface expression [19]. Transfected cells were kept in antibiotic selection with neomycin for two weeks, after which cells were tested in flow cytometry. The cells 1B7-RBL-2H3-HexB-CCR6h had 69% c-myc and 99% NGFR expression (Figure 3b).

The cells were assayed in a functional calcium mobilization assay, based on HexB cells activity, incubating the 1B7-RBL-2H3-HexB-CCR6h cells with serial dilutions of CCL20 to measure specific signal of CCR6 receptor (Figure 4a). 1B7-RBL-2H3-HexB cells without CCR6h were used as negative control. A concentration-response curve was realized showing an EC_50_ of 10.72 nM (Figure 4b).

### 3.3. Optimization and Performance of HTS Assay

Various experimental conditions were tested to optimize the assay for high-throughput screening (HTS). We found that the times of incubation tested (60, 70, 80 and 90 min) did not affect the EC_50_ value of the ligand CCL20 and the S/B ratio reached a plateau at 60 min of incubation (Figure 5a,b). Cells dispensing methods were examined by running the calcium mobilization assay with electronic multichannel pipette or manually plated cells. The EC_50_ value was the double when the cells were dispensed with electronic pipette and the signal-to-background ratio was better than the ratio obtained with manually plated cells (Figure 5c,d). Therefore, the final assay conditions for HTS were determined as follows: the time of ligand incubation was 60 min and the cells were dispensed with electronic pipette.

To see if this cell-based assay was suitable for HTS, Z’ factor and signal window were calculated. Using the fully optimized assay, we generated Z’-factor and signal window of assay running 48 positive controls and 48 negative controls. The average of the positive controls was 2164 RFU with a standard deviation of 107.1 RFU and the average of the negative controls was 848 RF with a standard deviation of 54.9 RFU. As shown in Figure 6a, the Z’ factor was 0.63 and the signal window 7.75. A Z’-factor ≥ 0.5 and a signal window ≥ 2 are appropriate for HTS assay; therefore, our assay is deemed to be highly robust and reproducible, and hence suitable for HTS applications.

Furthermore, to investigate reproducibility between plates, the corresponding wells from two plates were treated with the same concentration of CCL20 and the data were investigated with linear regression analysis. The correlation coefficient was 0.95, showing a high degree of reproducibility between duplicate plates (Figure 6b).

## 4. Discussion

GPCRs represent the largest family of druggable targets due to their numerous physiological and pathological roles, together with their potential for therapeutic intervention via using small molecules as regulators [1]. An important focus of the pharmaceutical industry is the development of robust, reliable, nonradioactive, homogenous, cost effective and easily adapted to a microtiter plate format (96-, 384-, or 1536-well) for robotic automation assays [22]. Although in the past decades signaling-dependent cell-based functional assays have been made, biased GPCR signaling complicates drug discovery efforts. The development of biased ligands as therapeutics heralds an era of increased drug efficacy with reduced drug side effects [23].

The chemokine receptor CCR6 is a GPCR that plays an important role in immunity and is involved in numerous diseases such as inflammatory bowel disease, psoriasis, multiple sclerosis and rheumatoid arthritis [24]. This GPCR is also involved in non-autoimmune diseases such as cancer or atherosclerosis. Thus, inhibition of CCR6 signaling would be an attractive strategy for the treatment of various diseases and a small molecule inhibitor of CCR6 could be useful for both in vitro and in vivo pathophysiological studies [25]. In this manner, the availability of a simple, rapid and robust assay to monitor CCR6h activity would expedite the search for drugs against CCR6.

There are some functional assays to screen compounds that bind to CCR6 receptor based on different biological measurement corresponding to different signaling pathways. Commercially available CCR6 assays include those that measure of cAMP accumulation (cAMP Hunter™ eXpress assay), flash increment of Ca^2+^ by FLIPR technology or β-arrestin recruitment (PathHunter^®^ eXpress CCR6 assay) in different cells: CHO-K1, cell lines like Chem-1 with endogenous high levels of promiscuous G_α15_ that promotes coupling G_o/i_ receptors to an increase in intracellular calcium and CHO-K1/U2OS cell lines, respectively. But to the date, around sixteen CCR6-CCL20 inhibitors have been investigated and no therapeutic agents targeting CCR6 have progressed into clinical evaluation [12]; therefore, it seems to be necessary to develop reliable new technologies for screening of CCR6h.

β-hexosaminidase (HexB) assay is useful for detecting mast cell degranulation upon a short stimulation with drugs or chemical agents [26]. This protein is used as reporter for granule secretion but has been traditionally considered a low sensitivity reporter with a low signal-to-background ratio and a strong variability between experiments due to a large variation over time in the amount of enzyme stored in the granules [27].

In the present investigation, β-hexosaminidase was overexpressed inside the granules of hematopoietic cells, RBL-2H3, reaching an improved signal-to-background ratio assay and a stable production of HexB by transfected cells. While untransfected cells have strong variability, cells transfected with HexB gene were more stable. Thus, such mast cells overexpressing HexB become sensitive cell-based sensors with low variability to measure exocytosis.

To ensure that 1B7-RBL-2H3-HexB cells expressed CCR6 receptor on their surface, the receptor was cloned into the cells. In these cells, when CCL20 activates CCR6 there is a release of HexB, encoded by the gene that was previously overexpressed as describe above, due to the increase in intracellular calcium.

Calcium mediated degranulation of RBL-2H3 cells is induced by GPCR signaling pathways and is associated with Ca^2+^ mobilization [28]. The release of hexosaminidase contained in the granules was detected by a substrate that measures extracellular β-hexosaminidase activity due to the activation of CCR6 by CCL20 ligand calcium signaling. The untransfected cells 1B7-RBL-2H3-HexB, without expression of CCR6 in a natural manner similar to their parental cell RBL-2H3, had a very low level of hexosaminidase secretion under ligand stimulation, demonstrating that the release of hexosaminidase in transfected cells is specific of CCL20 activation. Therefore, production of Ca^2+^ in this reporter cell line, the subsequent releasing of HexB and the generation of fluorescent signal can be attributed to the activation of a particular signaling pathway, initiated by the activation of calcium mobilization signaling of CCR6 receptor by CCL20 ligand. Then, like IgE receptor, at least some GPCRs induce degranulation upon activation in RBL-2H3.

The VGS sequence has been reported [19] to enhance surface expression of GPCRs and, therefore, was used to promote CCR6 expression on the plasma membrane. In the sTable 1B7-RBL-2H3-HexB-CCR6 cells the response to CCL20 was potent and had a robust signal with an EC_50_ of 10.72 nM.

The assay was then optimized, and it tested the time of ligand incubation and the method of cell dispensing. The results indicate that the assay performance is well within 60 min of CCL20 incubation and electronic cell dispensing.

With the optimized conditions, we calculated some parameters commonly used as tools to measure assay performance during validation of an assay for HTS. The Z’ factor is a useful tool used to assess the robustness of assay for screening [20] and the signal window (SW) provides a degree of separation between signals [21]. Although both parameters measure that assay signal adjusted for assay variability, is described that Z’ has better precision properties than SW to measure assay performance [29]. Nevertheless, we calculated both parameters for a better support to the method developed. In general, a Z’ value greater than or equal than 0.5 suggests that separation band is large and that the assay is robust enough for HTS and a signal window above 2 is recommended. Therefore, the assay was deemed robust and reproducible with a Z’ value of 0.63 and a SW value of 7.75. Moreover, the correlation of two plates of assay confirms the excellent reproducibility.

An ideal GPCR screening technology should be simple, nonradioactive, with high signal-to-noise ratio, homogeneous, with minimal reagents additions and be amenable to a microtiter plate format to facilitate robotic automation [16]. We used the HexB stored inside granules of hematopoietic cells to detect CCL20 induced exocytosis on CCR6 overexpressing cells and obtain an assay with a high signal-to-background ratio and low inter-assay variability. The assay performed very well, had a low baseline response, was reproducible and was suitable for high throughput screens. The sensor of the present investigation is faster than sensors based on inducible promoters, no lysis is needed for release of reporters, no washing or stop steps are needed thus increasing throughput, is stable and high signal to background is obtained for a robust assay with low variability between inter-assay experiments.

In theory, this calcium mobilization assay could be used to identify other ligands as well, including antagonist, inverse agonist or allosteric modulators, although further studies are needed to confirm this hypothesis. Moreover, these applications could be expanded to other GPCRs, provided their activation can also be coupled to an increase in calcium release.

Thus, we established a cell-based HTS assay system for monitoring calcium mobilization due to CCR6h activation by CCL20 using β-hexosaminidase activity. This technology represents a valuable tool in screening, counter screening and SAR characterization of compounds that modulate CCR6 activity.

## 5. Conclusions

In the present report, we generated a cell-based calcium mobilization assay for CCR6h receptor. The calcium mobilization assay established here, using measurement of β-hexosaminidase induced by the activation of 1B7-RBL-2H3-HexB-CCR6 cells calcium signaling by CCL20, can be reliably used to screen and study modulators of CCR6h.

This assay exhibited good values of Z’ and signal window and- thus can be applied in high-throughput screening formats. It has been demonstrated that the assay described here is robust, time saving and cost effective, generating highly reproducible results. Moreover, the assay could be easily extended to study modulators of other GPCRs. In conclusion, our approach provides an alternative strategy for ligand screening of CCR6h.

## Figures and Tables

**Figure 1 biomedicines-10-00422-f001:**
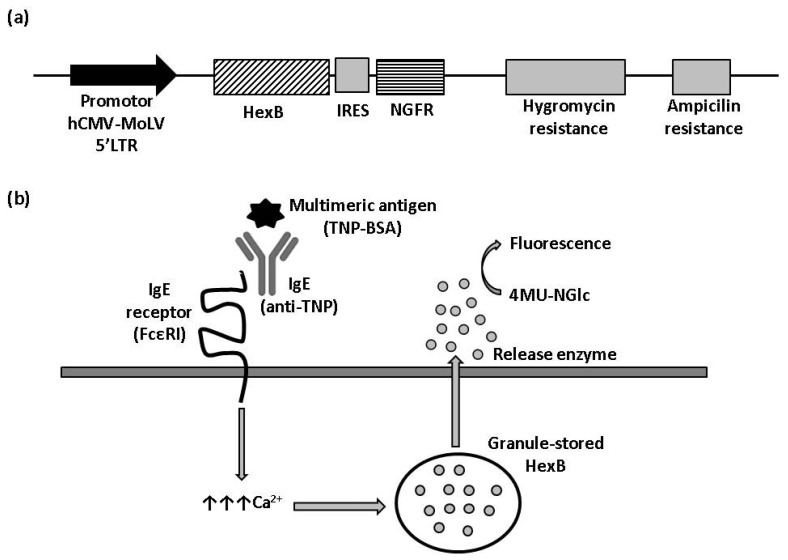
β-hexosaminidase cell sensor-based assay. (**a**) Vector construction. Plasmid vector with hygromycin resistance cassette used to stably express HexB fused to NGFR under control hCMV-MoLV-5′LTR promoter. (**b**) Illustration of HexB-based calcium mobilization assay. 4MU-NGlc: 4-Methylumbelliferyl N-acetyl-β-D-glucosaminide. The arrows represent an increment of calcium.

**Figure 2 biomedicines-10-00422-f002:**
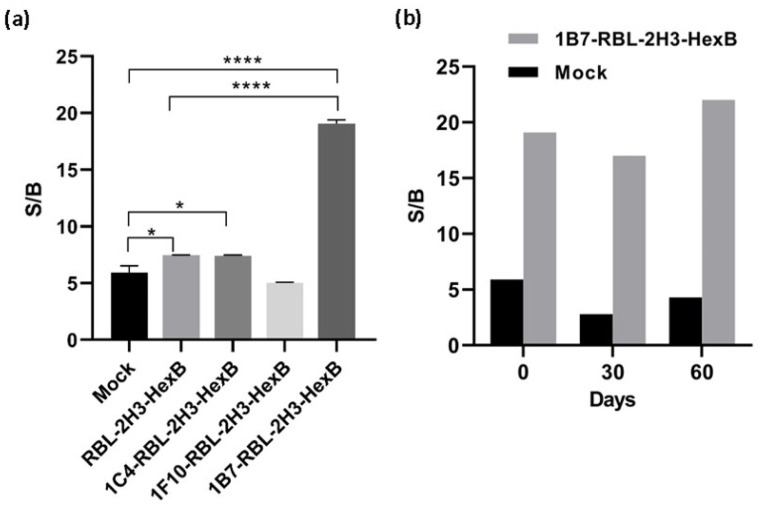
Activity of hexosaminidase released by transfected cells. (**a**) Signal-to-background ratio of mock transfectants, RBL-2H3 cells transfected with HexB and RBL-2H3-HexB clones. Means and SD of triplicates are plotted. (**b**) Stability of 1B7-RBL-2H3-HexB cells activity in the time. Representative data for one experiment are shown. * *p* < 0.1, **** *p* < 0.0001.

**Figure 3 biomedicines-10-00422-f003:**
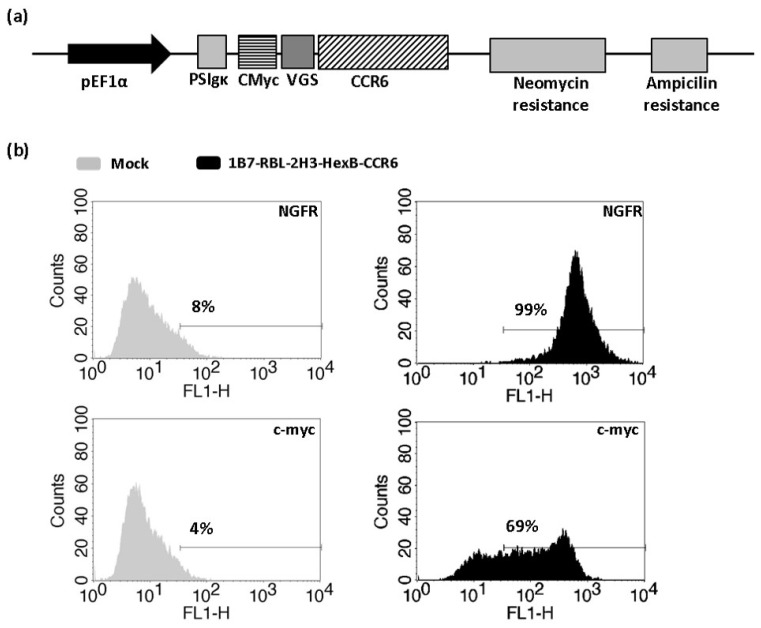
Obtaining of 1B7-RBL-2H3-HexB-CCR6h cells. (**a**) Vector construction. Plasmid vector with neomycin resistance cassette used to stably express CCR6h fused to c-myc on 1B7-RBL-2H3-HexB cells. (**b**) Flow cytometry. Expression level of stably transfected cells with HexB and CCR6h. One representative analysis from three independent experiments is shown. The lines represent the marker of the positive signal.

**Figure 4 biomedicines-10-00422-f004:**
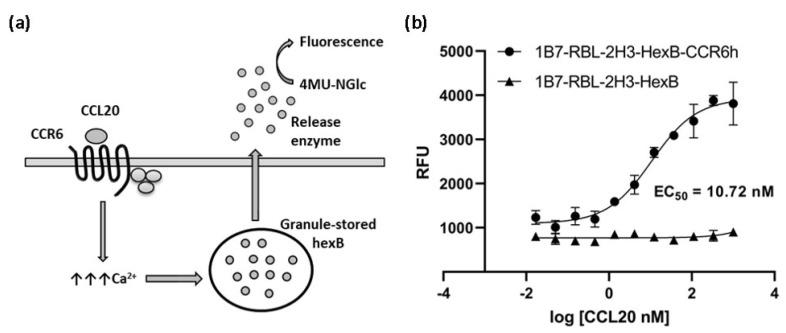
Calcium mobilization assay based on cell sensor. (**a**) Illustration of calcium mobilization assay for CCR6h. The arrows represent an increment of calcium. (**b**) Concentration-response curve of CCL20 in 1B7-RBL-2H3-HexB-CCR6h cells and 1B7-RBL-2H3-HexB cells. Means and SD of triplicates are plotted. RFU: relative fluorescence units.

**Figure 5 biomedicines-10-00422-f005:**
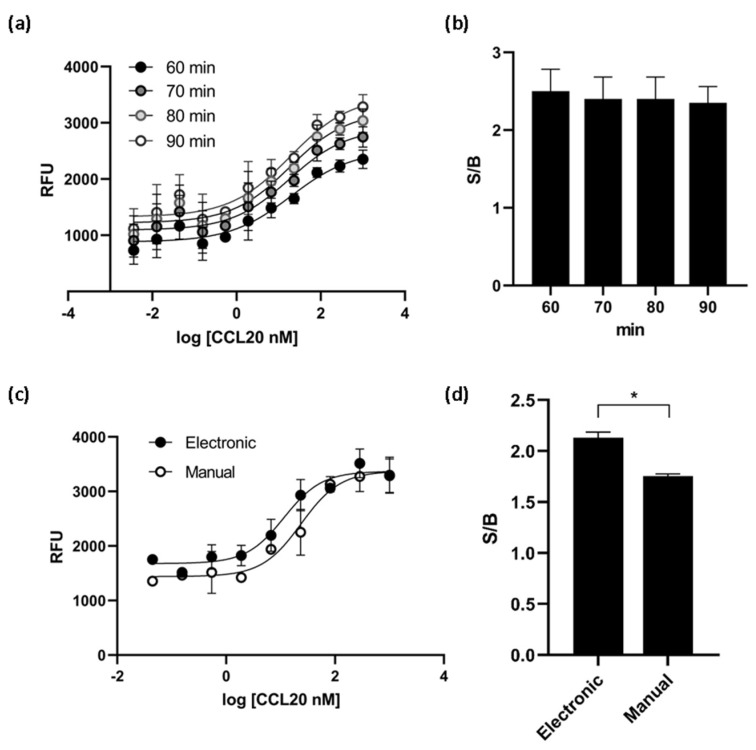
CCR6h calcium mobilization assay optimization. At different incubation times, the EC_50_ values obtained (**a**) and the S/B ratio (**b**) were almost the same. The cell dispensing method affect to dose response curve (**c**) and the S/B ratio was significantly higher with electronic method (**d**). * *p* < 0.1. Means and SD of triplicates are plotted. RFU: relative fluorescence units.

**Figure 6 biomedicines-10-00422-f006:**
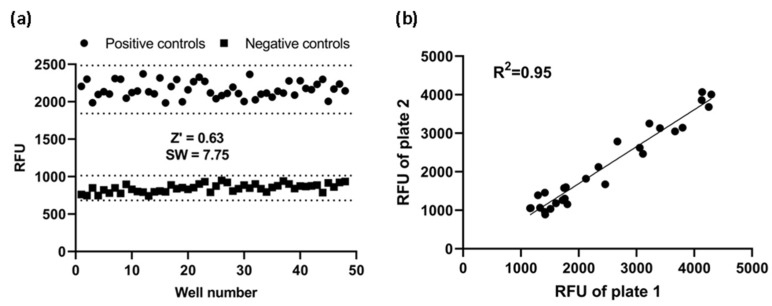
Assay performance. (**a**) Z’ factor and signal window (SW) determination. At the fully optimized assay, 48 replicates of positive and negative controls were studied. Dotted lines indicate mean ± 3SD of 48 data points. (**b**) Reproducibility. The same concentration of CCL20 was added to corresponding wells of two plates. Reproducibility of data from duplicates plates was investigated with a linear regression analysis. RFU: relative fluorescence units.

**Table 1 biomedicines-10-00422-t001:** β-hexosaminidase activity in different cell lines after 60 min of incubation.

Cell	Specific Exocytosis (RFU)	Background Release (RFU)
Mock	6424 ± 584	2056 ± 30
RBL-2H3-HexB	14,149 ± 309	2908 ± 77
1C4-RBL-2H3-HexB	39,185 ±1024	6302 ± 132
1F10-RBL-2H3-HexB	17,378 ± 362	4394 ± 83
1B7-RBL-2H3-HexB	28,554 ± 790	2611 ± 61
Without cells (blank)	1203 ± 42	1176 ± 47

RFU, relative fluorescence units.

## Data Availability

The data that support the findings of this study are available from the corresponding author upon reasonable request.

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
