# Peer review of "Development of a High-Throughput Calcium Mobilization Assay for CCR6 Receptor Coupled to Hydrolase Activity Readout"

_biomedicines, 2022, doi:10.3390/biomedicines10020422_

Round 1

Reviewer 1 Report

This manuscript is describing about development of high-throughput calcium mobilization assay for CCR6 receptor coupled to ß-hexosaminidase activity. Intracellular calcium mobilization of transfected cells was measured with a fluorescent substrate using activity of released the enzyme activity. This method was found to be robust with interesting data and to be used for screening of CCR6 modulators. This manuscript is well written and it is recommended to be acceptable.

Author Response

Reviewer: This manuscript is describing about development of high-throughput calcium mobilization assay for CCR6 receptor coupled to ß-hexosaminidase activity. Intracellular calcium mobilization of transfected cells was measured with a fluorescent substrate using activity of released the enzyme activity. This method was found to be robust with interesting data and to be used for screening of CCR6 modulators. This manuscript is well written and it is recommended to be acceptable.

Authors: We thank the reviewer for the careful reading of the manuscript and the kind comments. 

Reviewer 2 Report

The topic is very important. CCL6 receptor drives many harmful processes in the human organism.

However , I think that paper can be improved. Authors used a Z score for the assessment of the test value. However, the better way to assess test utility would be through the assessment of its power, sensitivity, specificity and NPV, PPV.

Author Response

Reviewer: The topic is very important. CCL6 receptor drives many harmful processes in the human organism.

Authors: We thank the reviewer for your kind comments. 

Reviewer: However, I think that paper can be improved. Authors used a Z score for the assessment of the test value. However, the better way to assess test utility would be through the assessment of its power, sensitivity, specificity and NPV, PPV.

Authors: We agree with you that power, sensitivity, specificity and NPV, PPV are good parameters to assess general test utility. However, Z′-factor is the gold standard to assess the robustness and performance of an assay to be used in high throughput screening that is the goal of the described cell-based assay [1], [2].  For this reason we used this score to evaluate the robustness and the quality of the assay developed in the manuscript.

  1. JH, C. TD, and O. KR, “A Simple Statistical Parameter for Use in Evaluation and Validation of High Throughput Screening Assays,” J. Biomol. Screen., vol. 4, no. 2, pp. 67–73, 1999, doi: 10.1177/108705719900400206.
  2. PW, E. BJ, S. GS, and C. KL, “A comparison of assay performance measures in screening assays: signal window, Z’ factor, and assay variability ratio,” J. Biomol. Screen., vol. 11, no. 3, pp. 247–252, Apr. 2006, doi: 10.1177/1087057105285610.

Reviewer 3 Report

41,42 line - Authors should write the exact mechanism of calcium mobilization by trimeric G proteins including the Gbetagamma subunit and the Galfa(q) subunit. Galfa(i) subunit reduces cAMP production and does not affect on Ca mobilization. Galfa(i) protein contains Gbetagamma subunit and the Galfa(i) subunit

Authors should show whether the transfected CCR6 and hexB proteins are expressed e.g. by western blot.

Authors must demonstrate that CCR6 activation causes calcium mobilization. The authors must demonstrate that CCR6-dependent calcium mobilization causes the release of hexB. It is possible that CCR6 causes the release of hexB through a pathway unrelated to calcium mobilization.

Author Response

Reviewer: 41,42 line - Authors should write the exact mechanism of calcium mobilization by trimeric G proteins including the Gbetagamma subunit and the Galfa(q) subunit. Galfa(i) subunit reduces cAMP production and does not affect on Ca mobilization. Galfa(i) protein contains Gbetagamma subunit and the Galfa(i) subunit

Authors: We agree with this suggestion and several information has been added.

In lines 35-44 the following information has been added: “GPCR receptors do not have activity per se but they are linked to GTPases that participate in signal transduction. Such GTPases are collectively known as G proteins and each G protein comprises three subunits, α, β, and γ. There are four families of G proteins that are classified according to their α subunit: Gs, Gi, Gq and G12/13 where Gs and Gi participate in cAMP-dependent pathways, Gq activates phospholipase C-dependent pathways and G12/13 activates a small family of GTPases [3-5]. In addition, the Gq family is further divided into four Gq proteins due to the use of four different α polypeptides, αq, α11, α14, and α15/16 [6]. The Gα15/16 polypeptide has a restricted tissue expression and is predominantly expressed in myeloid and B-cell lineages while Gαq and Gα11 are ubiquitously expressed [6]”.

In lines 50-53 the sentence in question has been changed in accordance with the suggestion of the reviewer. The sentence has been modified in the manuscript in this way: “Calcium mobilization is the result of activation of GPCRs through specific subunit Gαq activating classical phospholipase C pathways resulting in the release of calcium from the endoplasmatic reticulum into the cytoplasm [7]”.

Finally in lines 69-82 the following information has been added: “Extensive research has been done about the molecular determinants of coupling of each GPCR to a particular family of G proteins and today it is known that GPCR and G protein coupling selectivity involves the interaction of GPCR with both the extreme carboxyl terminus and linker I of Gα protein [15] and that development of chimeras of Gαq comprising the C-terminal five amino of Gαi or Gαs and a highly conserved glycine within Linker I is enough for coupling of non-Gαq coupled GPCR to signalling through phospholipase C-dependent pathways [16]. In addition, it was previously known that Gα15/16 is a promiscuous G protein, that is, it can couple to a wide variety of GPCR to signal through phospholipase C-dependent pathways [17] and this is the molecular basis of measurement of calcium release upon activation of many non-naturally Gαq coupled-GPCRs assays that are used in HTS today, for example, the ChemiScreen™ family of HTS products. RBL-2H3 is a rat basophilic leukemia cell line that expresses the Gα15/16 of Gα subunit. This promiscuous Gα15/16 protein shifts the receptors coupling to the calcium mobilization pathway [18]”

All the changes have been appropriately marked with the “track changes function”.

Reviewer: Authors should show whether the transfected CCR6 and hexB proteins are expressed e.g. by western blot.

Authors: We agree that it is important to measure expression of transfected proteins and in our work the expression level of proteins in transfected cells is evaluated by flow cytometry of the tags included in the plasmids (Figure 3b).

To show the CCR6h receptor expression on surface of cells, the cells were transfected with an expression plasmid containing a c-myc tag fused in frame upstream of CCR6h (Figure 3a) and flow cytometry of c-myc was tested (figure 3b).

To show HexB protein expression on cells, the cells were transfected with a plasmid for the expression of human HexB fused to IRES-NGFR cassette (Figure 1A) and flow cytometry of NGFR was tested (figure 3b). This is a bicistronic expression plasmid and the expression of NGFR indicates that HexB is also expressed. Moreover, the activity of HexB protein in transfected cells was tested by fluorescence assay (Table1).

Reviewer: Authors must demonstrate that CCR6 activation causes calcium mobilization. The authors must demonstrate that CCR6-dependent calcium mobilization causes the release of hexB. It is possible that CCR6 causes the release of hexB through a pathway unrelated to calcium mobilization.

Authors: We agree with you, this is a very interesting issue. There are two lines of evidence that support our work and conclusions:

1) It is known that hematopoietic cells, like RBL-2H3, express the family Gα15/16 of Gα subunit. The promiscuous Gα15/16 protein shifts the receptors coupling to the calcium mobilization pathway [1].

2) A second line of evidence is that it has been described that Ca2+-mediated degranulation of RBL-2H3 cells is induced by GPCR signaling pathways and is associated with Ca2+ mobilization [2].

  1. Zhu, L. Fang, and X. Xie, “Development of a universal high-throughput calcium assay for G-protein-coupled receptors with promiscuous G-protein Gα15/16,” Acta Pharmacol. Sin. 2008 294, vol. 29, no. 4, pp. 507–516, Apr. 2008, doi: 10.1111/j.1745-7254.2008.00775.x.
  2. YC et al., “Differential Ca2+ mobilization and mast cell degranulation by FcεRI- and GPCR-mediated signaling,” Cell Calcium, vol. 67, pp. 31–39, Nov. 2017, doi: 10.1016/J.CECA.2017.08.002

In fact, there are companies that commercialize RBL-2H3 transfected with CCR6 to measure calcium release upon ligand binding, for example, the Millipore´s Ready-to-Assay CCR6 kit (Cat. No. HTS011RTA) or Eurofins ChemiSCREEN™ CCR6 cell-based assay (Cat. No. HTS011C).

This has been clarified in the revised paper with additional information as described before.

Finally, we have added the following sentence in the Discussion: “Calcium mediated degranulation of RBL-2H3 cells is induced by GPCR signaling pathways, and is associated with Ca2+ mobilization [29].”

All the changes have been appropriately marked with the “track changes function”.

Round 2

Reviewer 3 Report

I agree that calcium mobilization causes degranulation.

I agree that GPCR causes calcium mobilization.

However, it may be an misinterpretation that GPCR causes degranulation through calcium mobilization. GPCR can cause degranulation through calcium mobilization. However, GPCR not only causes calcium mobilization but also activates other pathways. Gbetagamma also activates PI3K, which causes degranulation [1]. GPCR can also act independently of G proteins [2].

Authors must show that CCR6 causes calcium mobilization, and that this calcium mobilization causes the release of HexB.

[1] Nanamori M, Chen J, Du X, Ye RD. Regulation of leukocyte degranulation by cGMP-dependent protein kinase and phosphoinositide 3-kinase: potential roles in phosphorylation of target membrane SNARE complex proteins in rat mast cells. J Immunol. 2007 Jan 1;178(1):416-27. doi: 10.4049/jimmunol.178.1.416.

[2] Neel NF, Sai J, Ham AJ, Sobolik-Delmaire T, Mernaugh RL, Richmond A. IQGAP1 is a novel CXCR2-interacting protein and essential component of the "chemosynapse". PLoS One. 2011;6(8):e23813. doi: 10.1371/journal.pone.0023813.

This manuscript is a resubmission of an earlier submission. The following is a list of the peer review reports and author responses from that submission.